# Quasi-equilibrium polariton condensates in the non-linear regime and beyond

Ned Goodman[1] , Brendan C. Mulkerin[1,2] , Jesper Levinsen[1,2] and Meera M. Parish[1,2]

**1** School of Physics and Astronomy, Monash University, Victoria 3800, Australia
**2** ARC Centre of Excellence in Future Low-Energy Electronics Technologies, Monash University, Victoria 3800, Australia

## Abstract

We investigate the many-body behavior of polaritons formed from electron-hole pairs strongly coupled to photons in a two-dimensional semiconductor microcavity. We use a microscopic mean-field BCS theory that describes polariton condensation in quasi-equilibrium across the full range of excitation densities. In the limit of vanishing density, we show that our theory recovers the exact single-particle properties of polaritons, while at low densities it captures non-linear polariton-polariton interactions within the Born approximation. For the case of highly screened contact interactions between charge carriers, we obtain analytic expressions for the equation of state of the many-body system. This allows us to show that there is a photon resonance at a chemical potential higher than the photon cavity energy, where the electron-hole pair correlations in the polariton condensate become universal and independent of the details of the carrier interactions. Comparing the effect of different ranged interactions between charge carriers, we find that the Rytova-Keldysh potential (relevant to transition metal dichalcogenides) offers the best prospect of reaching the BCS regime, where pairs strongly overlap and the minimum pairing gap occurs at finite momentum. Finally, going beyond thermal equilibrium, we argue that there are generically two polariton branches in the driven-dissipative system and we discuss the possibility of a density-driven exceptional point within our model.


doi:10.21468/SciPostPhys.15.3.116

# 1   Introduction

When light is confined in a semiconductor microcavity, it can become strongly coupled to excitons (bound electron-hole pairs) to form hybrid light-matter quasiparticles — exciton polaritons [1–3]. Such polaritons inherit properties of both light and matter, thus providing a versatile platform for exploring a range of quantum phenomena such as Bose-Einstein condensation (BEC) and superfluidity at elevated temperatures [4,5], polaron-polaritons in the presence of charge doping [6], and non-Hermitian topological effects [7]. One particular regime of interest is that of large excitation densities, where the interparticle spacing starts to approach the exciton Bohr radius $a_0$ and non-linear interaction effects play a dominant role [8,9]. Here, there is the prospect of achieving a crossover from a polariton BEC to a BCS-like state analogous to the paired state in superconductors [10–13]. However, there is yet to be an unambiguous observation of the BEC-BCS crossover in current polariton experiments [14–18]. In particular, a measurement of a BCS-like pairing gap at high densities remains elusive.

Theoretically, it is challenging to describe the high-density regime of the exciton-polariton system since it is a complex many-body problem where the microscopic composite nature of the excitons must be included. This goes beyond the usual coupled-oscillator description of exciton polaritons which treats the exciton as a structureless bosonic mode [2]. For the case of localized excitons (e.g., due to disorder), it is possible to study the effect of the underlying electrons and holes by approximating each electron-hole pair as a two-level system, corresponding to a generalization of the Dicke model [3,19–21]. However, this cannot capture the possibility of overlapping electron-hole pairs and related BCS pairing phenomena, which require the relative motion of electrons and holes.

To describe the mobile case, previous theoretical works have employed a BCS variational wave function, which accounts for the internal structure of electron-hole pairs within a mean-field approximation [10–13, 22]. Such an approach has provided important insight into the BEC-BCS crossover in cold-atomic and excitonic systems in the absence of coupling to light [23–25]. However, an outstanding question in the polariton system is the effect of high-momentum unbound electron-hole pairs, which are present in the microscopic model and which have been shown to modify the cavity photon [26], but which have so far been ignored in the mean-field theories. Furthermore, there remain questions about how the many-body electron-hole-photon description is connected to the polariton BEC at low densities, with Ref. [22] obtaining polariton-polariton interactions from the BCS mean-field theory that are weaker than expected based on few-body calculations [27].

In this work, we resolve these questions regarding the polariton BEC-BCS crossover by employing a microscopic mean-field theory that properly renormalizes the high-energy electron-hole pairs using exact few-body calculations [26, 28]. We formally show that the BCS mean-field approach recovers the expected properties of a polariton BEC at low densities, and we find that the polariton-polariton interaction strength agrees with that obtained within the standard Born approximation [26, 27]. We furthermore compare different types of interactions between charge carriers, including the dielectrically screened Rytova-Keldysh potential which appears in atomically thin transition-metal dichalcogenides (TMDs) [29]. In particular, we find that, due to bandgap renormalization, the Rytova-Keldysh potential in TMDs offers the best prospect of reaching the BCS regime of strongly overlapping electron-hole pairs, where there is a BCS-like pairing gap at finite momentum.

With increasing excitation density, we show that the condensate ground state eventually hits a photon resonance at a chemical potential that lies above the cavity photon energy. Here we find that the system becomes photon dominated and that the electron-hole correlations become universal and independent of the range of the carrier interactions. In the case of strongly screened contact interactions between carriers, we can go further and derive an analytic expression for the electron-hole-photon equation of state for any excitation density. This allows us to extend our results for the equilibrium ground state to the driven-dissipative non-equilibrium system, where we investigate the possibility of an exceptional point involving upper and lower polariton branches [30, 31].

The paper is organized as follows. In Sec. 2 we set out the theoretical model and renormalization schemes for the long-range Coulomb and Rytova-Keldysh interactions and short-range contact interaction. In Sec. 3 we present the general BCS variational formalism for the equilibrium system, which allows us to investigate coherent phenomena across the full range of excitation densities and which provides a benchmark for more complex non-equilibrium theories. Section 4 presents the analytical results for the case of screened contact interactions, while Sec. 5 discusses the general behaviour of the BCS-BEC crossover and presents our numerical results for the case of long-range interactions. Finally, in Sec. 6 we discuss the driven dissipative system and its connection to the many-body upper and lower branches. We conclude in Sec. 7.

## 2 Model and few-body properties

We consider a two-dimensional (2D) semiconductor embedded in a planar microcavity, such that photons can excite electron-hole pairs across the semiconductor band gap. This scenario can be modeled with an effective low-energy Hamiltonian that includes electrons, holes, and photons [3]

$$
\begin{aligned}
\hat{H} = &\sum_{\mathbf{k}}\left(\epsilon_{\mathbf{k}}^{e}e_{\mathbf{k}}^{\dagger}e_{\mathbf{k}}+\epsilon_{\mathbf{k}}^{h}h_{\mathbf{k}}^{\dagger}h_{\mathbf{k}}\right)+\sum_{\mathbf{k}}(\omega_{0}+\epsilon_{\mathbf{k}}^{c})c_{\mathbf{k}}^{\dagger}c_{\mathbf{k}} \\
&-\frac{1}{2}\sum_{\mathbf{kk'q}}V_{\mathbf{q}}\left(2e_{\mathbf{k+q}}^{\dagger}h_{\mathbf{k'-q}}^{\dagger}h_{\mathbf{k'}}e_{\mathbf{k}}-e_{\mathbf{k+q}}^{\dagger}e_{\mathbf{k'-q}}^{\dagger}e_{\mathbf{k'}}e_{\mathbf{k}}-h_{\mathbf{k+q}}^{\dagger}h_{\mathbf{k'-q}}^{\dagger}h_{\mathbf{k'}}h_{\mathbf{k}}\right) \\
&+g\sum_{\mathbf{kq}}\left(e_{\mathbf{k}}^{\dagger}h_{\mathbf{q-k}}^{\dagger}c_{\mathbf{q}}+c_{\mathbf{q}}^{\dagger}h_{\mathbf{q-k}}e_{\mathbf{k}}\right).
\end{aligned}
\tag{1}
$$

Here, $c_{\mathbf{k}}^{\dagger}$, $e_{\mathbf{k}}^{\dagger}$, and $h_{\mathbf{k}}^{\dagger}$ respectively create cavity photons, electrons, and holes with in-plane momentum $\mathbf{k}$, while the corresponding 2D dispersions are $\epsilon_{\mathbf{k}}^{\alpha}=|\mathbf{k}|^{2}/2m_{\alpha}\equiv k^{2}/2m_{\alpha}$ in terms of the effective masses of the photons, electrons, and holes, $m_{\mathrm{c}}$, $m_{\mathrm{e}}$, and $m_{\mathrm{h}}$, respectively. For convenience, we write the cavity photon frequency at zero momentum, $\omega_{0}$, separately, and we

measure all energies from the band gap energy. We also neglect the spin degrees of freedom, since we are interested in the simplest minimal model for exciton-polariton condensation. Note that throughout this paper we work in units where $\hbar$ and the system area $\mathcal{A}$ are both 1.

The potential $V_{\mathbf{q}}$ corresponds to the interactions between charge carriers. In this work, we consider the long-range Coulomb and the Rytova-Keldysh potentials, which are typically employed for semiconductor quantum wells and atomically thin TMDs, respectively. In both cases, the interaction originates from the three-dimensional Coulomb interaction, and the difference between the two potentials arises from the different dielectric environments in the two geometries [32–34]. For comparison, we also consider the case of a highly screened short-range contact interaction. As we will show, the latter has the advantage that it admits a semi-analytical solution for the relevant thermodynamic properties, and therefore it acts as a highly useful benchmark for other theories.

Finally, the light-matter interactions are parameterized by the bare coupling strength $g$, which we take to be constant up to an ultraviolet (UV) momentum cutoff $\Lambda$. We have chosen a form of light-matter coupling where only $s$-orbital electron-hole states couple to the photon, and we have utilized the rotating wave approximation, which is reasonable when the semiconductor band gap greatly exceeds all other energy scales in the problem.

## 2.1 Renormalization scheme

Experimentally, the parameters characterizing the light-matter coupled system are typically determined by comparing the measured optical spectrum (e.g., absorption) with the expected energy eigenvalues for two coupled oscillators (excitons and photons in this case) [9],

$$E_{\pm} = -\varepsilon_{\mathrm{B}} + \frac{1}{2}\left(\delta \pm \sqrt{\delta^2 + 4\Omega^2}\right). \tag{2}$$

Here, $\pm$ refers to the upper and lower polaritons, respectively, $\varepsilon_{\mathrm{B}}$ is the exciton binding energy, and the energies are measured from the electron-hole band gap. Both the effective cavity photon-exciton detuning $\delta$ and the light-matter Rabi coupling $\Omega$ can be obtained from a fit to the polariton energies at low excitation density.

In a similar manner, we can theoretically obtain the physical parameters for a single polariton by comparing the spectrum calculated within the microscopic Hamiltonian (1) with Eq. (2). This allows us to relate the physical observables, the photon-exciton detuning $\delta$ and Rabi coupling $\Omega$, to the bare parameters of the model, i.e., $\omega_0$, $g$, and the UV cutoff $\Lambda$. Given the need for a UV cutoff, this procedure formally involves the process of renormalization [26]. The precise identification depends on the form of the electronic interactions, and this has previously been performed for Coulomb interactions [26], the Rytova-Keldysh potential [35], and for the case of strongly screened contact interactions [28, 36]. This procedure has, for instance, allowed the accurate simulation of experimentally observed [37] diamagnetic shifts in the presence of both a strong magnetic field and strong light-matter coupling [38]. For completeness, in the remainder of this section we briefly summarize these renormalization schemes below.

### 2.1.1 Long-range interactions

We start by considering the case of interactions between charge carriers that scale as $1/r$ at large interparticle separation $r$, appropriate for either quantum wells or atomically thin semiconductors in the microcavity. In the absence of light-matter coupling, the most general state for an electron-hole pair is

$$|\Phi\rangle = \sum_{\mathbf{k}} \phi_{\mathbf{k}} e_{\mathbf{k}}^{\dagger} h_{-\mathbf{k}}^{\dagger} |0\rangle, \tag{3}$$

where $\phi_{\mathbf{k}}$ is the bare exciton wave function, and we have the normalization condition $\langle \Phi | \Phi \rangle = \sum_{\mathbf{k}} |\phi_{\mathbf{k}}|^2 = 1$. The state $|0\rangle$ denotes the electron-hole-photon vacuum. The wave function $\phi_{\mathbf{k}}$ satisfies the Schrödinger equation:

$$(E - \bar{\epsilon}_{\mathbf{k}})\phi_{\mathbf{k}} = -\sum_{\mathbf{k}'} V_{\mathbf{k}-\mathbf{k}'}\phi_{\mathbf{k}'} , \tag{4}$$

where $\bar{\epsilon}_{\mathbf{k}} = \epsilon_{\mathbf{k}}^e + \epsilon_{\mathbf{k}}^h = k^2/2m_{\mathrm{r}}$ corresponds to the total kinetic energy and we write the electron-hole reduced mass, $m_{\mathrm{r}} = (1/m_{\mathrm{e}} + 1/m_{\mathrm{h}})^{-1}$. The negative-energy solutions of this equation correspond to the exciton bound states; in this work, we focus on the lowest energy $s$-wave ($1s$) state with binding energy $\varepsilon_{\mathrm{B}}$.

In semiconductor quantum wells, the interactions between electrons and holes are typically described by the long-range 2D Coulomb potential:

$$V_{\mathbf{q}} = \frac{\pi}{m_{\mathrm{r}}a_0 q} , \tag{5}$$

where $a_0$ is the effective 2D Bohr radius. In this case, Eq. (4) yields negative energy solutions corresponding to the infinite hydrogenic series of exciton bound states. In particular, the $1s$ bound state has the wave function

$$\phi_{\mathbf{k}} = \frac{\sqrt{8\pi}a_0}{(1 + k^2 a_0^2)^{3/2}} , \tag{6}$$

binding energy $\varepsilon_{\mathrm{B}} = 1/2m_{\mathrm{r}}a_0^2$, and associated Rabi coupling $\Omega = g\sqrt{2/\pi}/a_0$.

For atomically thin semiconductors, the bare Coulomb interaction is modified by dielectric screening at short distances, giving the Rytova-Keldysh potential [32–34]

$$V_{\mathbf{q}}^{\mathrm{RK}} = \frac{\pi}{m_{\mathrm{r}}a_0 q}\frac{1}{1 + r_0 q} , \tag{7}$$

where $r_0$ is the effective screening length which is typically of the order $r_0 = 1 \sim 10\mathrm{nm}$ [39,40]. In the absence of coupling to light, the $1s$ exciton binding energy and wave function must be solved numerically via Eq. (4), and are functions of the screening length $r_0$, i.e., $\varepsilon_{\mathrm{B}}(r_0)$ and $\phi_{\mathbf{k}}(r_0)$. To highlight the dependence on the screening length, we plot in Fig. 1 the $1s$ binding energy $\varepsilon_{\mathrm{B}}(r_0)$ in units of the $r_0 = 0$ solution, as a function of $r_0/a_0$. We see that as the screening length increases, the exciton binding energy decreases with respect to $\varepsilon_{\mathrm{B}}(r_0 = 0)$. However, note that the binding energy in TMDs is typically much larger than that in semiconductor quantum wells since the Bohr radius $a_0$ in Eq. (7) can be orders of magnitude smaller than the one in Eq. (5) [29].

Due to the choice of a short-range electron-hole-photon interaction, the bare coupling $g$ leads to an arbitrarily large shift of the cavity photon frequency which should be renormalized, as shown in Refs. [26,35]. To this end, we take the most general electron-hole-photon superposition

$$|\Psi\rangle = \sum_{\mathbf{k}} \varphi_{\mathbf{k}} e_{\mathbf{k}}^{\dagger} h_{-\mathbf{k}}^{\dagger} |0\rangle + \gamma c_0^{\dagger} |0\rangle , \tag{8}$$

where $\varphi_{\mathbf{k}}$ and $\gamma$ are the exciton and photonic wave functions, respectively. We ensure the total wave function is normalized according to $\langle \Psi | \Psi \rangle = \sum_{\mathbf{k}} |\varphi_{\mathbf{k}}|^2 + |\gamma|^2 = 1$. Taking the Schrödinger equation at energy $E$ and projecting it onto the electron-hole and photon subspaces, $\langle 0| e_{\mathbf{k}} h_{-\mathbf{k}} (\hat{H} - E) |\Psi\rangle = 0$ and $\langle 0| c_0 (\hat{H} - E) |\Psi\rangle = 0$, gives [26]

$$(E - \bar{\epsilon}_{\mathbf{k}})\varphi_{\mathbf{k}} = -\sum_{\mathbf{k}'} V_{\mathbf{k}-\mathbf{k}'}\varphi_{\mathbf{k}'} + g\gamma , \tag{9a}$$

$$(E - \omega_0)\gamma = g\sum_{\mathbf{k}} \varphi_{\mathbf{k}} . \tag{9b}$$

Inserting Eq. (9a) into Eq. (9b) and rearranging, yields

$$\left( E - \omega_0 + g^2 \sum_{\mathbf{k}} \frac{1}{\bar{\epsilon}_{\mathbf{k}} - E} \right) \gamma = g \sum_{\mathbf{k}\mathbf{k}'} \frac{V_{\mathbf{k}-\mathbf{k}'} \varphi_{\mathbf{k}'}}{\bar{\epsilon}_{\mathbf{k}} - E} \, . \tag{10}$$

In the case of the long-range Coulombic potentials considered here, the sum on the right-hand side of Eq. (10) is convergent for $k \rightarrow \infty$. However, the sum on the left-hand side is logarithmically divergent and depends on the UV momentum cutoff $\Lambda$. To obtain finite cutoff-independent results when light-matter coupling is present, we require the bare cavity frequency $\omega_0$ to cancel the logarithmic divergence, leading to the renormalized frequency

$$\omega = \omega_0 - g^2 \sum_{\mathbf{k}} \frac{1}{\bar{\epsilon}_{\mathbf{k}} + \varepsilon_{\mathrm{B}}} \, . \tag{11}$$

Here, we have assumed that the photon is resonant with the $1s$ exciton such that the energy $E \simeq -\varepsilon_{\mathrm{B}}$, which is valid to logarithmic accuracy. Comparing to the coupled-oscillator approximation for the upper and lower polariton energies for a weakly light-matter coupled system in Eq. (2), we obtain the renormalized photon-exciton detuning and Rabi coupling as [26]

$$\delta = \omega + \varepsilon_{\mathrm{B}} \, , \tag{12}$$

$$\Omega = g \sum_{\mathbf{k}} \phi_{\mathbf{k}} \, , \tag{13}$$

respectively, valid for both long-range potentials. With this identification, Eq. (9) yields solutions that, for $\Omega \ll \varepsilon_{\mathrm{B}}$, are well described by the model of two coupled oscillators, Eq. (2), whereas there are some corrections in the regime of very strong light-matter coupling where $\Omega \sim \varepsilon_{\mathrm{B}}$ [26].

The Hopfield coefficients (light and matter amplitudes $C$ and $X$) can also be calculated numerically from Eq. (9) using the fact that the photonic Hopfield coefficient is given by the variational parameter $\gamma$ together with the normalization condition: we have $|C_{\pm}|^2 = |\gamma_{\pm}|^2$ and $|X_{\pm}|^2 = 1 - |C_{\pm}|^2$, where $\pm$ again refers to upper and lower polaritons energies found from numerically solving Eq. (9).

### 2.1.2 Short-range interactions

When the interactions between electrons and holes are strongly screened,[1] the long-range Coulomb potential can be approximated by a short-range contact interaction, i.e., a constant potential in momentum space, $V_{\mathbf{q}} = V_0 > 0$. It is well known that the contact interaction needs to be renormalized, and within light-matter coupled systems this has been carried out previously in several works: [28, 36, 41, 42]. Here, we utilize the scheme of Ref. [28], which uses the same cutoff for the carrier interaction as for the light-matter coupling, since this results in a significant simplification of the renormalization scheme and has the advantage of being fully analytic. This approach is reasonable since both the coupling to light and the exciton binding rely on the behavior of the electron-hole wave function at short distances, and thus both cutoffs are expected to be well-approximated by the inverse lattice spacing. Importantly, the low-energy physics that we aim to describe is independent of the precise manner in which the cutoffs are introduced.

The contact interaction admits only a single electron-hole bound state, with $V_0$ related to the exciton binding energy $\varepsilon_{\mathrm{B}}$ via Eq. (4) evaluated at $E = -\varepsilon_{\mathrm{B}}$:

$$(\varepsilon_{\mathrm{B}} + \bar{\epsilon}_{\mathbf{k}}) \phi_{\mathbf{k}} = V_0 \sum_{\mathbf{k}'} \phi_{\mathbf{k}'} \, . \tag{14}$$

---

[1] Note that the Thomas-Fermi screening due to mobile charges is qualitatively different from the dielectric screening in the Rytova-Keldysh potential, since it affects the Coulomb interaction at small $q$ rather than large $q$.

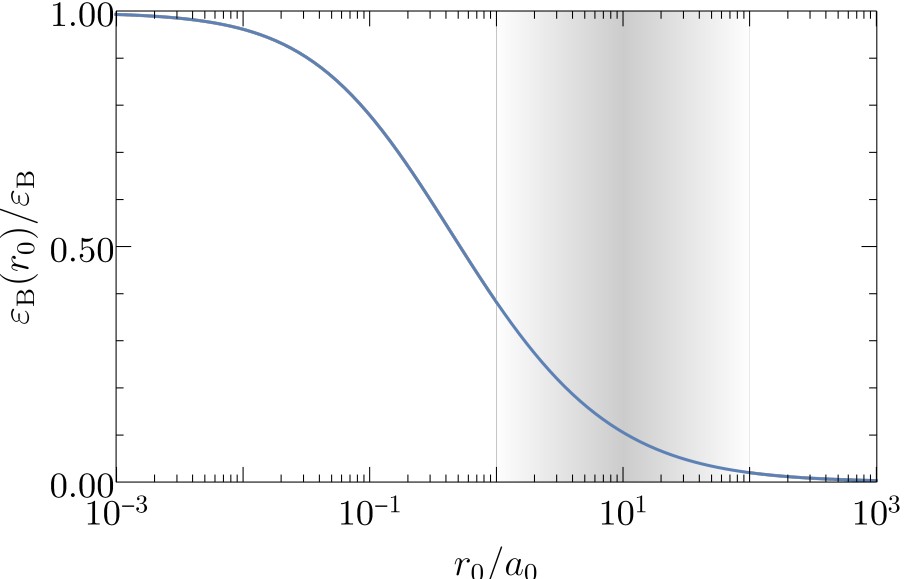

Figure 1: The 1s exciton binding energy $\varepsilon_B(r_0)$ in units of the zero screening (Coulomb) binding energy as a function of the dielectric screening length. The typical range of $r_0/a_0$ values in TMD materials is depicted by the shaded region.

This can be rearranged to yield

$$\frac{1}{V_0} = \sum_{\mathbf{k}}^{\Lambda} \frac{1}{\varepsilon_B + \bar{\epsilon}_{\mathbf{k}}} \,. \tag{15}$$

We explicitly see that the bare coupling $V_0$ vanishes logarithmically as the momentum cutoff $\Lambda \to \infty$. The corresponding wave function is [28]

$$\phi_{\mathbf{k}} = \sqrt{\frac{2\pi\varepsilon_B}{m_r}} \frac{1}{\varepsilon_B + \bar{\epsilon}_{\mathbf{k}}} \,, \tag{16}$$

where we define an effective Bohr radius $a_0 = 1/\sqrt{2m_r \varepsilon_B}$.

To obtain the polariton spectrum using the short-range contact interaction, and relate the bare parameters to physical quantities, we consider the general light-matter state (8) and use the contact interactions $V_{\mathbf{q}} = V_0$ in Eq. (9). The upper and lower polariton energies can then be obtained from the negative-energy solutions of the implicit equation [28],

$$(\omega_0 - E)\ln\left(\frac{-E}{\varepsilon_B}\right) = \frac{\Omega^2}{\varepsilon_B} \,, \tag{17}$$

where the effective Rabi coupling for the contact potential is

$$\Omega = g\sum_{\mathbf{k}} \phi_{\mathbf{k}} = \frac{g}{V_0}\sqrt{\frac{2\pi\varepsilon_B}{m_r}} \,. \tag{18}$$

To identify the detuning, we consider $\Omega \ll \varepsilon_B$ and expand around $E = -\varepsilon_B$. Comparing with the expected polariton energies in Eq. (2) then yields [28]

$$\delta = \underbrace{\omega_0 - \frac{\Omega^2}{2\varepsilon_B}}_{\omega} + \varepsilon_B \,. \tag{19}$$

Here the bare cavity frequency $\omega_0$ is independent of the cutoff $\Lambda$, in contrast to the case of Coulomb interactions in Eq. (11). Instead, it is the bare coupling $g$ that vanishes logarithmically as $\Lambda \to \infty$ similarly to $V_0$, as can be seen from Eq. (18).

Finally, one can also obtain the Hopfield coefficients analytically [28], giving

$$|C_\pm|^2 \equiv |\gamma_\pm|^2 = \frac{1}{1 + \frac{\varepsilon_B}{|E_\pm|} \frac{(E_\pm - \omega_0)^2}{\Omega^2}} \,, \tag{20}$$

and $|X_\pm|^2 = 1 - |C_\pm|^2$, where $E_\pm$ are the polariton solutions of Eq. (17).

## 3 BCS approach

We now apply the model and renormalization schemes to the scenario of the many-body problem consisting of electrons, holes, and photons in the semiconductor microcavity. To investigate many-body coherent phenomena, we focus on the equilibrium system at zero temperature and consider a mean-field BCS-like variational wave function [10, 11]:

$$|\Psi_{\text{BCS}}\rangle = e^{\lambda c_0^\dagger - \lambda c_0} \prod_{\mathbf{k}} \left( u_{\mathbf{k}} + v_{\mathbf{k}} e_{\mathbf{k}}^\dagger h_{-\mathbf{k}}^\dagger \right) |0\rangle \,, \tag{21}$$

where we can take the variational parameters $(u_{\mathbf{k}}, v_{\mathbf{k}}, \lambda)$ to be real without loss of generality. This wave function combines a BCS ansatz for electron-hole pairs with a coherent state of photons, such that the overall phase is well defined but the number of excitations in the microcavity is uncertain. For the wave function to be normalized we require $u_{\mathbf{k}}^2 + v_{\mathbf{k}}^2 = 1$.

We obtain the ground-state properties through the free energy, $F = \langle \Psi_{\text{BCS}} | \hat{K} | \Psi_{\text{BCS}} \rangle$, where $\hat{K} = \hat{H} - \mu \hat{N}_{\text{tot}}$. In terms of the variational parameters $u_{\mathbf{k}}$, $v_{\mathbf{k}}$, and $\lambda$, the free energy is given by

$$F = \sum_{\mathbf{k}} (\bar{\epsilon}_{\mathbf{k}} - \mu) v_{\mathbf{k}}^2 + (\omega_0 - \mu) \lambda^2 + 2g\lambda \sum_{\mathbf{k}} u_{\mathbf{k}} v_{\mathbf{k}} - \sum_{\mathbf{k} \neq \mathbf{k}'} V_{\mathbf{k} - \mathbf{k}'} u_{\mathbf{k}} v_{\mathbf{k}} u_{\mathbf{k}'} v_{\mathbf{k}'} - \sum_{\mathbf{k} \neq \mathbf{k}'} V_{\mathbf{k} - \mathbf{k}'} v_{\mathbf{k}}^2 v_{\mathbf{k}'}^2 \,. \tag{22}$$

Here, $\hat{N}_{\text{tot}} = \sum_{\mathbf{k}} \left[ c_{\mathbf{k}}^\dagger c_{\mathbf{k}} + \frac{1}{2} (e_{\mathbf{k}}^\dagger e_{\mathbf{k}} + h_{\mathbf{k}}^\dagger h_{\mathbf{k}}) \right]$ is the total number of (bosonic) excitations and $\mu$ is the associated chemical potential. Within the BCS ansatz (21), the photon density is given by

$$n_c = \sum_{\mathbf{k}} \langle c_{\mathbf{k}}^\dagger c_{\mathbf{k}} \rangle = \langle c_0^\dagger c_0 \rangle = \lambda^2 \,, \tag{23}$$

while the electron and hole densities are (assuming charge neutrality)

$$n_e = n_h = \sum_{\mathbf{k}} \langle e_{\mathbf{k}}^\dagger e_{\mathbf{k}} \rangle = \sum_{\mathbf{k}} v_{\mathbf{k}}^2 \,. \tag{24}$$

The total excitation density is then $n_{\text{tot}} = n_c + n_e$.

In the absence of any interactions, the system decouples into non-interacting photons and charge carriers. In this case, a finite density of electrons (or holes) forms a Fermi sea with Fermi wave vector $k_F = (4\pi n_e)^{1/2}$. We will use $k_F$ as a measure of the charge carrier density in general.

To determine the variational parameters $u_{\mathbf{k}}$, $v_{\mathbf{k}}$, and $\lambda$, we minimize the free energy by defining $u_{\mathbf{k}} = \cos\theta_{\mathbf{k}}$ and $v_{\mathbf{k}} = \sin\theta_{\mathbf{k}}$, and then taking the stationary conditions $\partial F / \partial \theta_{\mathbf{k}} = 0$ and $\partial F / \partial \lambda = 0$. This yields the two coupled equations:

$$\left( \bar{\epsilon}_{\mathbf{k}} - \mu - 2 \sum_{\mathbf{k}'} V_{\mathbf{k} - \mathbf{k}'} v_{\mathbf{k}'}^2 \right) u_{\mathbf{k}} v_{\mathbf{k}} + \left[ u_{\mathbf{k}}^2 - v_{\mathbf{k}}^2 \right] \left( g\lambda - \sum_{\mathbf{k}'} V_{\mathbf{k} - \mathbf{k}'} u_{\mathbf{k}'} v_{\mathbf{k}'} \right) = 0 \,, \tag{25a}$$

$$(\omega_0 - \mu)\lambda + g \sum_{\mathbf{k}} u_{\mathbf{k}} v_{\mathbf{k}} = 0 \,. \tag{25b}$$

In the low-density limit where $v_{\mathbf{k}} \ll 1$ and $u_{\mathbf{k}} \to 1$ [24], Eq. (25) reduces to

$$(\bar{\epsilon}_{\mathbf{k}} - \mu)v_{\mathbf{k}} + g\lambda - \sum_{\mathbf{k}'} V_{\mathbf{k}-\mathbf{k}'} v_{\mathbf{k}'} = 0 \,, \tag{26a}$$

$$(\omega_0 - \mu)\lambda + g\sum_{\mathbf{k}} v_{\mathbf{k}} = 0 \,, \tag{26b}$$

which is equivalent to the set of equations for a single polariton, Eq. (9), once we identify $\mu$ with the polariton energy, $\lambda \approx \sqrt{n_{\mathrm{tot}}}\gamma$, and $v_{\mathbf{k}} \approx \sqrt{n_{\mathrm{tot}}}\varphi_{\mathbf{k}}$. Thus, we expect $\mu \to E_-$ and $n_{\mathrm{c}}/n_{\mathrm{tot}} \to |\gamma_-|^2$ as $n_{\mathrm{tot}} \to 0$ in the zero-temperature ground state. Note that there is also an excited-state solution that is connected to the upper polariton at low densities, which we discuss in Sec. 6.

To solve Eq. (25) for arbitrary density, we follow the standard BCS approach [43] (see also Ref. [10]) and define an order parameter

$$\Delta_{\mathbf{k}} = -g\lambda + \sum_{\mathbf{k}'} V_{\mathbf{k}-\mathbf{k}'} u_{\mathbf{k}'} v_{\mathbf{k}'} \,, \tag{27}$$

as well as the modified single-particle dispersion

$$\xi_{\mathbf{k}} = \frac{1}{2}(\bar{\epsilon}_{\mathbf{k}} - \mu) - \sum_{\mathbf{k}'} V_{\mathbf{k}-\mathbf{k}'} v_{\mathbf{k}'}^2 \,. \tag{28}$$

Here, the interaction-dependent term in Eq. (28) can be viewed as a form of bandgap renormalization within the BCS ansatz. From Eq. (25a), we then obtain

$$\frac{u_{\mathbf{k}} v_{\mathbf{k}}}{u_{\mathbf{k}}^2 - v_{\mathbf{k}}^2} = \frac{1}{2}\tan 2\theta_{\mathbf{k}} = \frac{\Delta_{\mathbf{k}}}{2\xi_{\mathbf{k}}} \,. \tag{29}$$

Using trigonometric identities, the coupled equations in Eq. (25) finally become

$$\Delta_{\mathbf{k}} = -g\lambda + \sum_{\mathbf{k}'} V_{\mathbf{k}-\mathbf{k}'} \frac{\Delta_{\mathbf{k}'}}{2\sqrt{\xi_{\mathbf{k}'}^2 + \Delta_{\mathbf{k}'}^2}} \,, \tag{30a}$$

$$\lambda = -\frac{g}{\omega_0 - \mu} \sum_{\mathbf{k}} \frac{\Delta_{\mathbf{k}}}{2\sqrt{\xi_{\mathbf{k}}^2 + \Delta_{\mathbf{k}}^2}} \,. \tag{30b}$$

Equation (30a) is a BCS-like gap equation for the order parameter $\Delta_{\mathbf{k}}$ while Eq. (30b) describes the photon field amplitude $\lambda$. In addition, we have the electron-hole momentum occupation

$$v_{\mathbf{k}}^2 = \frac{1}{2}(1 - \cos 2\theta_{\mathbf{k}}) = \frac{1}{2}\left(1 - \frac{\xi_{\mathbf{k}}}{\sqrt{\xi_{\mathbf{k}}^2 + \Delta_{\mathbf{k}}^2}}\right) \,, \tag{31}$$

which allows us to determine the electron (hole) density $n_{\mathrm{e}}$ and the single-particle energy $\xi_{\mathbf{k}}$.

In general, one must solve the set of equations (30) numerically by iteration, from which all other quantities such as the Bogoliubov dispersion and electron and hole densities follow. Importantly, while Eq. (30) depends on the bare parameters of the model, these should be related to the physical parameters of the exciton-polariton spectrum as discussed in Sec. 2.1. For the case of long-range Coulomb and Rytova-Keldysh potentials, we arrange Eq. (30b) into a renormalized form by substituting (30a) into (30b) and using (11) to finally give

$$\lambda\left[\omega - \mu - g^2 \sum_{\mathbf{k}}\left(\frac{1}{2\sqrt{\xi_{\mathbf{k}}^2 + \Delta_{\mathbf{k}}^2}} - \frac{1}{\bar{\epsilon}_{\mathbf{k}} + \varepsilon_{\mathrm{B}}}\right)\right] = -g\sum_{\mathbf{k} \neq \mathbf{k}'} \frac{V_{\mathbf{k}-\mathbf{k}'}\Delta_{\mathbf{k}'}}{4\sqrt{\xi_{\mathbf{k}}^2 + \Delta_{\mathbf{k}}^2}\sqrt{\xi_{\mathbf{k}'}^2 + \Delta_{\mathbf{k}'}^2}} \,, \tag{32}$$

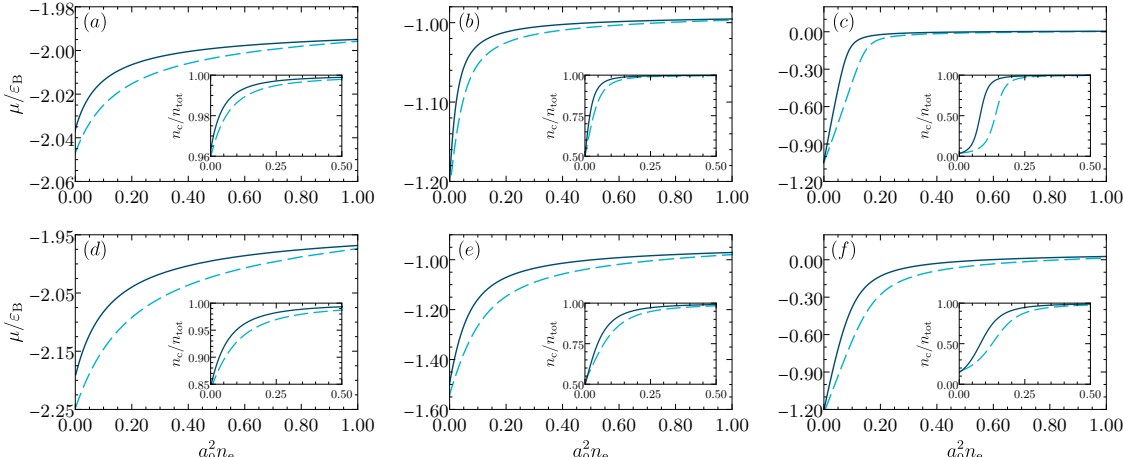

Figure 2: Chemical potential of the electron-hole-photon ground state as a function of electron density for short-range (solid) and Coulomb (dashed) interactions. The Rabi couplings are (a,b,c) $\Omega/\varepsilon_{\mathrm{B}} = 0.2$ and (d,e,f) $\Omega/\varepsilon_{\mathrm{B}} = 0.5$, while the photon-exciton detuning increases from left to right: (a,d) $\delta/\varepsilon_{\mathrm{B}} = -1$, (b,e) $\delta/\varepsilon_{\mathrm{B}} = 0$, and (c,f) $\delta/\varepsilon_{\mathrm{B}} = 1$ . The insets show the corresponding photon fractions as a function of electron density.

where $g$ is related to the Rabi coupling via Eq. (13).

We note that the equations here only depend on the reduced mass $m_r$, which is a consequence of only considering pairs at zero momentum. However, going beyond the mean-field approximation, we expect the behavior to also involve the electron-hole mass ratio as well as the photon mass.

Finally, to consider the quasiparticle excitations, we note that the BCS-like wave function (21) is the vacuum for the Bogoliubov excitations $\gamma_{\mathbf{k}\uparrow} = u_{\mathbf{k}}e_{\mathbf{k}} - v_{\mathbf{k}}h_{-\mathbf{k}}^{\dagger}$ and $\gamma_{-\mathbf{k}\downarrow} = v_{\mathbf{k}}e_{\mathbf{k}}^{\dagger} + u_{\mathbf{k}}h_{-\mathbf{k}}$ as in standard BCS theory [43]. Therefore, at momentum $\mathbf{k}$ the pair-breaking energy of the two fermionic quasiparticles is $2E_{\mathbf{k}}$ with [44]

$$
\begin{aligned}
E_{\mathbf{k}} &= \frac{1}{2} \langle \Psi_{\mathrm{BCS}} | (\gamma_{\mathbf{k}\uparrow}\gamma_{-\mathbf{k}\downarrow}\hat{K}\gamma_{-\mathbf{k}\downarrow}^{\dagger}\gamma_{\mathbf{k}\uparrow}^{\dagger} - \hat{K}) | \Psi_{\mathrm{BCS}} \rangle \\
&= \sqrt{\Delta_{\mathbf{k}}^2 + \xi_{\mathbf{k}}^2},
\end{aligned}
\tag{33}
$$

being the average quasiparticle energy. This has the same form as in the usual BCS theory, with the effect of the coupling to light incorporated into the gap and the single-particle dispersion.

## 4  Short-range interactions: analytical results

Remarkably, when the interactions between charges are short-range such that $V_{\mathbf{q}} = V_0$, it is possible to solve the problem analytically once we relate the bare parameters to physical observables. In this case, the bandgap renormalization term in Eq. (28) vanishes as the cutoff $\Lambda \to \infty$, since $V_0 \to 0$ while $\sum_{\mathbf{k}} v_{\mathbf{k}}^2$ remains finite. Thus, the single-particle dispersion simply becomes $\xi_{\mathbf{k}} = \frac{1}{2}(\bar{\epsilon}_{\mathbf{k}} - \mu)$. Furthermore, since the interaction is constant in momentum space, the right hand side of Eq. (30a) is independent of momentum, and hence the order parameter is constant, i.e., $\Delta_{\mathbf{k}} \equiv \Delta$. These properties allow us to simplify the coupled BCS equations:

substituting (30b) into (30a) and dividing by $V_0$ and $\Delta$ we have

$$\frac{1}{V_0} - \sum_{\mathbf{k}} \frac{1}{2\sqrt{\xi_{\mathbf{k}}^2 + \Delta^2}} = \frac{1}{\omega_0 - \mu} \frac{g^2}{V_0} \sum_{\mathbf{k}} \frac{1}{2\sqrt{\xi_{\mathbf{k}}^2 + \Delta^2}} \, . \tag{34}$$

Using the relation in Eq. (15), we can write the bare coupling constant $V_0$ in terms of the exciton binding energy to remove the dependence on the UV momentum cutoff $\Lambda$ on the left hand side of Eq. (34), i.e.,

$$\frac{1}{V_0} - \sum_{\mathbf{k}} \frac{1}{2\sqrt{\xi_{\mathbf{k}}^2 + \Delta^2}} = \frac{m_{\mathrm{r}}}{2\pi} \ln\left( \frac{\sqrt{4\Delta^2 + \mu^2} - \mu}{2\varepsilon_{\mathrm{B}}} \right) . \tag{35}$$

To remove the cutoff dependence on the right hand side of Eq. (34), we use the definition of the Rabi coupling in Eq. (18) together with the fact that $V_0 \sum_{\mathbf{k}} \frac{1}{2\sqrt{\xi_{\mathbf{k}}^2 + \Delta^2}} \to 1$ as $\Lambda \to \infty$. Thus, Eq. (34) finally becomes

$$\frac{m_{\mathrm{r}}}{2\pi} \ln\left( \frac{\sqrt{4\Delta^2 + \mu^2} - \mu}{2\varepsilon_{\mathrm{B}}} \right) = \frac{1}{\omega_0 - \mu} \frac{m_{\mathrm{r}}}{2\pi\varepsilon_{\mathrm{B}}} \Omega^2 \, . \tag{36}$$

This can be rearranged into an analytical expression for the order parameter:

$$\Delta^2 = \varepsilon_{\mathrm{B}} e^{\frac{\Omega^2}{(\omega_0 - \mu)\varepsilon_{\mathrm{B}}}} \left( \varepsilon_{\mathrm{B}} e^{\frac{\Omega^2}{(\omega_0 - \mu)\varepsilon_{\mathrm{B}}}} + \mu \right). \tag{37}$$

We can also find closed-form analytic expressions for the excitation densities. Using the expressions for $v_{\mathbf{k}}^2$ [see Eq. (31)] and $\Delta$, we obtain the electron density

$$n_{\mathrm{e}} = \frac{m_{\mathrm{r}}}{2\pi} \left( \varepsilon_{\mathrm{B}} e^{\frac{\Omega^2}{(\omega_0 - \mu)\varepsilon_{\mathrm{B}}}} + \mu \right). \tag{38}$$

Note that in the limit $n_{\mathrm{e}} \to 0$, we recover the implicit energy equation (17) for a single polariton, as expected. The photon density can similarly be found by substituting Eq. (37) into (30b). Using the definition of the Rabi coupling in Eq. (18) and the renormalization scheme where $V_0 \sum_{\mathbf{k}} \frac{1}{2\sqrt{\xi_{\mathbf{k}}^2 + \Delta^2}} \to 1$ as $\Lambda \to \infty$, we find the photon field amplitude

$$\lambda = -\frac{\Omega\Delta}{\omega_0 - \mu} \sqrt{\frac{m_{\mathrm{r}}}{2\pi\varepsilon_{\mathrm{B}}}} \, . \tag{39}$$

Equations (37–39) are key results of this work. They show analytically how the order parameter and the densities depend on the chemical potential and the semiconductor microcavity parameters, where the bare frequency $\omega_0$ is related to the photon-exciton detuning via Eq. (19). In particular, we see that there is a singular point at $\omega_0 = \mu$ where the system is resonant with the bare cavity frequency and the densities diverge, as is apparent in Fig. 2. Such behavior has also been obtained in previous theoretical works [10, 13]; however, our analytical calculations show that the divergence in density is exponential and that the position of the resonance, $\omega_0$, is slightly higher than the cavity frequency $\omega$ that would be extracted from low-density measurements.

For vanishing light-matter coupling, $\Omega \to 0$, we recover the mean-field results for the BEC-BCS crossover in a 2D Fermi gas, as first derived by Randeria *et al.* [45, 46]. In this case, one can solve for the order parameter and chemical potential in terms of electron density, giving

$$\Delta = \sqrt{2E_F \varepsilon_{\mathrm{B}}} , \qquad \mu = 2E_F - \varepsilon_{\mathrm{B}} . \tag{40}$$

Here we have defined the "Fermi energy" $E_F = \pi n_{\mathrm{e}} / m_{\mathrm{r}}$, since this corresponds to the actual Fermi energy of a non-interacting electron (or hole) gas when the electron and hole masses are equal. We clearly see from the chemical potential in Eq. (40) how the system smoothly evolves from a Bose gas of dimers to a weakly interacting BCS state with increasing electron density, in the absence of any coupling to light.

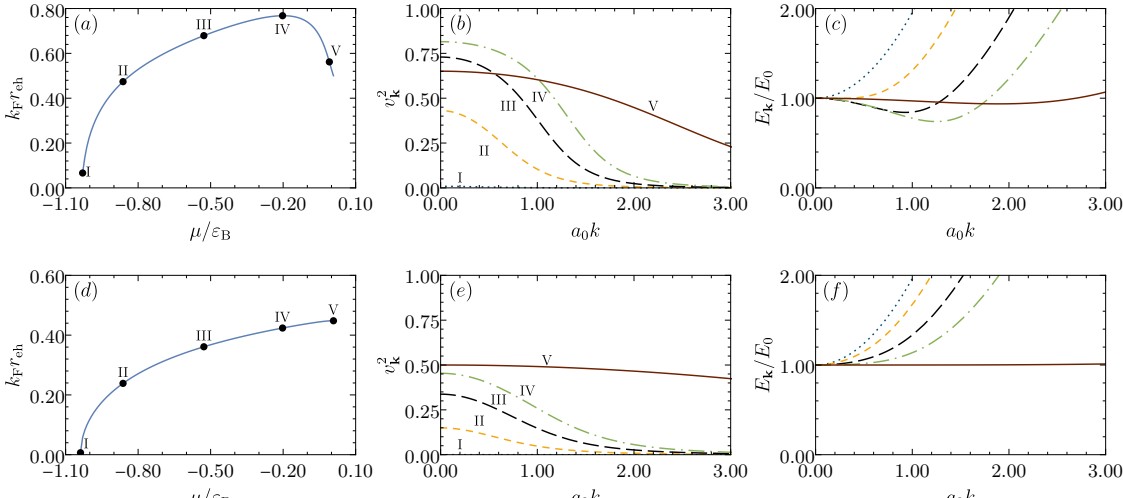

Figure 3: (a,d) Mean electron-hole pair size as a function of chemical potential for Rabi coupling $\Omega/\varepsilon_{\mathrm{B}} = 0.2$ and detuning $\delta/\varepsilon_{\mathrm{B}} = 1$. The corresponding momentum distributions (b,e) and quasiparticle excitation spectra (c,f) at the values of the chemical potentials indicated by the solid circles (I-V) in (a,d). The top and bottom panels are for Coulomb and contact interactions, respectively.

## 5  BEC-BCS crossover

We now turn to the behavior of the light-coupled system throughout the density-driven crossover for both short- and long-range interactions. Our results in these two scenarios are obtained, respectively, using the analytic expressions in Eqs. (37–39) or a numerical solution of Eq. (30). In the low-density limit, where there are tightly bound electron-hole pairs, we expect to recover a dilute Bose gas of exciton-polaritons. In this regime, the gas is well described as a weakly interacting Bose-Einstein condensate. Conversely, in the high-density regime, we expect the composite nature of the electron-hole pairs to become important, potentially leading to BCS-like pairing at the Fermi surface.

### 5.1  Low-density BEC regime

In the low density limit, the leading order contribution to the ground-state chemical potential is the lower polariton energy, as discussed in Sec. 3, while the next order term arises from interactions between polaritons. Thus, the low-density behavior of the ground state is governed by [47]

$$\mu = E_- + g_{\mathrm{PP}} n_{\mathrm{tot}}, \tag{41}$$

where $g_{\mathrm{PP}}$ is the polariton-polariton interaction strength which needs to be determined.

The calculation of $g_{\mathrm{PP}}$ is in general a complicated four-body problem, and it has only been performed in full for the case of short-range contact interactions [42]. Within the BCS ansatz (21), one can show that interactions are only captured within the Born approximation [26, 27, 48], such that we have (see Appendix A for a full derivation):

$$g_{\mathrm{PP}} = 2 \sum_{\mathbf{k}} (\bar{\epsilon}_{\mathbf{k}} - E_-) \varphi_{\mathrm{LP}\mathbf{k}}^4 - 2 \sum_{\mathbf{k}, \mathbf{k}'} V_{\mathbf{k}-\mathbf{k}'} \varphi_{\mathrm{LP}\mathbf{k}}^2 \varphi_{\mathrm{LP}\mathbf{k}'}^2, \tag{42}$$

where $\varphi_{\mathrm{LP}\mathbf{k}}$ is the electron-hole wave function of the lower polariton. The Born approximation provides an upper bound on the interaction strength between identical polaritons

and it is expected to become more accurate with increasing Rabi coupling [42]. For contact interactions, Eq. (42) can be evaluated analytically, giving $g_{PP} = 2\pi|X_-|^4/m_r$ with exciton fraction $|X_-|^2 = n_e/n_{tot}$. This turns out to be equivalent to taking $\varphi_{LPk} = |X_-|\phi_k$ in Eq. (42), with $\phi_k$ the exciton wave function in the absence of coupling to light. However, this identification is only approximately true for long-range interactions due to the light-induced changes to the electron-hole wave function [26,49]. For Coulomb interactions, Eq. (42) gives $g_{PP} \approx 6|X_-|^4\varepsilon_B a_0^2 = 3|X_-|^4/m_r$ [27], and there are some small deviations as $\Omega$ and $|\delta|$ approach $\varepsilon_B$ [26].

## 5.2 BEC-BCS crossover for strongly screened and Coulomb interactions

In Fig. 2 we compare the density dependence of the chemical potential for Coulomb and contact interactions at different detunings $\delta$ and Rabi couplings $\Omega$. Note that, at vanishing density, there is some difference in the values of the lower polariton energy when the detuning is large and negative [Fig. 2(a,d)], since the coupled-oscillator model (2) becomes less accurate far away from the exciton energy. We see that the initial increase (blueshift) of $\mu$ with density is steeper for the case of contact interactions, which is consistent with its larger polariton-polariton interaction strength (42). However, the behavior is qualitatively similar between short- and long-range interactions, with both featuring a resonance close to the cavity photon frequency where the chemical potential saturates and the densities diverge. In particular, the resonance in the Coulomb case lies slightly above $\omega = \delta - \varepsilon_B$, as in the contact case (see, also, Sec. 4). For all detunings, the system becomes photon dominated near the resonance (see insets in Fig. 2), even for positive detuning $\delta = 1$ where the condensate is largely excitonic at low densities.

The existence of the photon resonance also means that the ground state is confined to the region $\mu < 0$ for typical parameters in a microcavity, which is unlike the usual BCS-BEC crossover in the absence of light [24,45]. Furthermore, the crossover to the BCS regime is usually defined as the point where the excitation energy $E_k$ in Eq. (33) develops a minimum at finite momentum [50]. For the case of contact interactions, where $E_k = \sqrt{(\bar{\epsilon}_k - \mu)^2/4 + \Delta^2}$, this occurs when $\mu = 0$, which means that the BCS regime requires $\mu > 0$. Therefore, this raises questions about whether the BCS regime can be reached in the light-matter coupled system.

To further understand the nature of the BEC-BCS crossover in the polariton system, we consider different measures of the electron-hole pair correlations, as plotted in Fig. 3 for fixed Rabi coupling $\Omega/\varepsilon_B = 0.2$ and detuning $\delta/\varepsilon_B = 1$ [the same parameters as in Fig. 2(c)]. We estimate the size of the electron-hole pairs using the many-body wave function $\langle e_k^\dagger h_{-k}^\dagger \rangle$, defined in real space as

$$\Psi(\mathbf{r}) = \frac{\sum_k u_k v_k e^{i\mathbf{k}\cdot\mathbf{r}}}{\sqrt{\sum_k u_k^2 v_k^2}} \, . \tag{43}$$

Note that this reduces to the Fourier transform of the electron-hole wave function $\varphi_k$ for the polariton in the limit $v_k \to 0$. We then find the mean pair size via

$$r_{eh} = \int d\mathbf{r}\, r\, |\Psi(\mathbf{r})|^2 \, . \tag{44}$$

We compare this to the interparticle spacing (as encoded in the Fermi wave vector $k_F$) to determine how much the pairs overlap and thus how BCS-like the pairing is.

At low densities, $r_{eh}$ is roughly constant and given by the electron-hole separation in the lower polariton state. Thus, the evolution of $k_F r_{eh}$ versus chemical potential in Fig. 3 is initially determined by Eq. (41) such that $k_F r_{eh} \propto \sqrt{\mu - E_-}$, regardless of the range of the interactions.

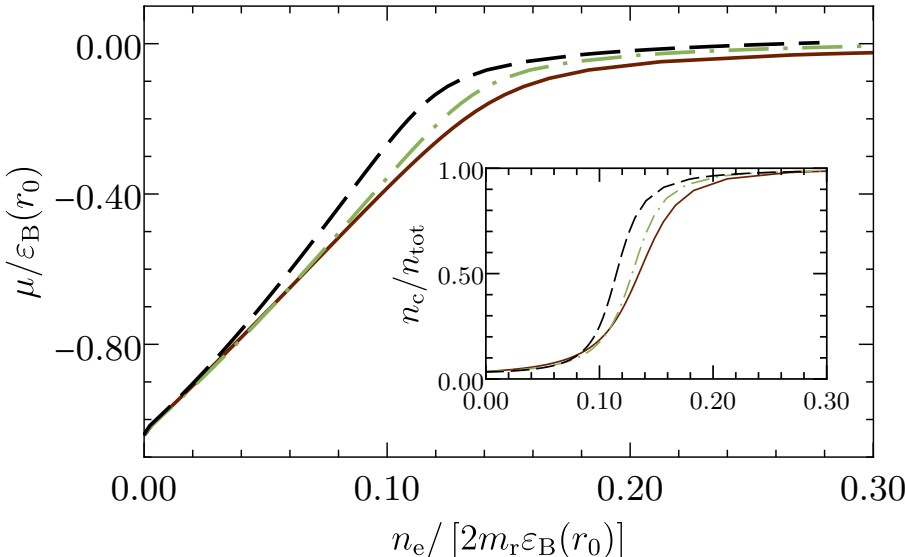

Figure 4: Ground-state chemical potential as a function of density for the Rytova-Keldysh potential at fixed Rabi coupling $\Omega(r_0)/\varepsilon_{\text{B}}(r_0) = 0.2$ and detuning $\delta(r_0)/\varepsilon_{\text{B}}(r_0) = 1$. The dielectric screening length is $r_0/a_0 = 0$, 1, and 10 (solid, dashed-dotted and dashed respectively). The inset shows the corresponding photon fraction as a function of density.

The corresponding excitation spectrum $E_{\mathbf{k}}$ in this regime is quadratic, with a minimum at $k = 0$, as expected for a condensate of tightly bound dimers.

With increasing density, the momentum distribution $v_{\mathbf{k}}^2$ smoothly evolves away from the electron-hole wave function $\varphi_{\mathbf{k}}^2$ for a single polariton and Pauli blocking plays a stronger role. For Coulomb interactions, the excitation dispersion in Fig. 3(c) develops a pronounced minimum at finite momentum, which shifts to larger values of momentum as the density increases. By contrast, the spectrum for contact interactions [Fig. 3(f)] displays no such BCS-like behavior even as the density approaches infinity. This difference in behavior can be traced back to the bandgap renormalization term in Eq. (28), which lowers the single-particle energies in the case of Coulomb interactions, resulting in a greater density for a given chemical potential. We can also see this in the behavior of $k_F r_{\text{eh}}$, which grows more steeply for Coulomb interactions, reaching a greater maximum value in Fig. 3(a).

Approaching the cavity resonance where the densities diverge, we see in Fig. 3(b,e) that the momentum distributions tend towards a constant value, $v_{\mathbf{k}} \to 1/2$, as expected from Eq. (31) when $\Delta_{\mathbf{k}} \to \infty$. Similarly, the scaled dispersion $E_{\mathbf{k}}/E_0$ flattens as it becomes dominated by the order parameter. Finally, the pair size $r_{\text{eh}}$ goes to zero (see, also, Refs. [10, 11]), but this does not result in a BEC of tightly bound electron-hole dimers. Rather, $r_{\text{eh}}$ becomes tied to the interparticle spacing which goes to zero as the density diverges. Using the fact that $\Delta_{\mathbf{k}}$ is dominated by the coupling to the cavity photon when $n_e, n_c \to \infty$, one can show that

$$\Delta_{\mathbf{k}} \simeq \frac{2\pi n_e}{m_r}, \qquad u_{\mathbf{k}} v_{\mathbf{k}} \simeq \frac{\Delta_{\mathbf{k}}}{\sqrt{\bar{\epsilon}_{\mathbf{k}}^2 + 4\Delta_{\mathbf{k}}^2}} . \tag{45}$$

This gives the universal result $k_F r_{\text{eh}} \simeq 0.448$ at resonance, which should hold for any type of matter interactions. For the case of contact interactions, this corresponds to the maximum

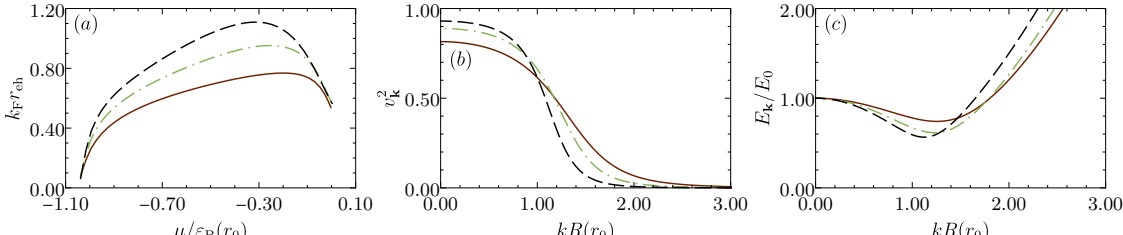

Figure 5: Electron-hole pair correlations throughout the BEC-BCS crossover for the Rytova-Keldysh potential, where we have used Rabi coupling $\Omega/\varepsilon_{\mathrm{B}}(r_0) = 0.2$, detuning $\delta/\varepsilon_{\mathrm{B}}(r_0) = 1$, and dielectric screening lengths $r_0/a_0 = 0$, 1, and 10 (solid, dashed-dotted and dashed respectively). We plot (a) the electron-hole pair size $k_F r_{\mathrm{eh}}$ as a function of chemical potential and the corresponding momentum distributions (b) and excitation spectrums (c) at the chemical potential where $k_F r_{\mathrm{eh}}$ is maximal, i.e., $\mu = -0.2, -0.25, -0.3$ for screening lengths $r_0/a_0 = 0$, 1, 10, respectively. Here we have defined the effective radius for the exciton as $R(r_0) = 1/\sqrt{2m_{\mathrm{r}}\varepsilon_{\mathrm{B}}(r_0)}$.

value of $k_F r_{\mathrm{eh}}$ [Fig. 3(d)], i.e., the point where pairs are maximally overlapping.

## 5.3 Crossover for Rytova-Keldysh interactions

We now turn to the effects of dielectric screening within the Rytova-Keldysh potential on the BEC-BCS crossover. We again focus on the case of excitonic detuning with $\delta(r_0)/\varepsilon_{\mathrm{B}}(r_0) = 1$ and $\Omega(r_0)/\varepsilon_{\mathrm{B}}(r_0) = 0.2$, where the parameters now depend on the additional lengthscale $r_0$ (see Fig. 1). As shown in Fig. 4, the evolution of the chemical potential with density is not significantly modified by screening. The low-density behavior is once again governed by Eq. (41), and the larger slope for $r_0/a_0 = 10$ indicates a larger interaction strength $g_{\mathrm{PP}}$ for the Rytova-Keldysh potential, which is consistent with calculations for exciton-exciton interactions within the Born approximation [51]. The larger $g_{\mathrm{PP}}$ also results in a greater photon fraction for a given electron density (inset of Fig. 4) since the chemical potential approaches the cavity resonance faster.

While the density dependence of the chemical potential appears to approach that of a screened short-range potential with increasing $r_0/a_0$, we find that the electron-hole pair correlations display a completely different evolution. Figure 5(a) shows that the maximum pair size grows with increasing $r_0/a_0$ and exceeds $1/k_F$ for $r_0/a_0 = 10$. This implies that the dielectrically screened system can go deeper into the BCS regime, an observation which is further supported by the more step-like behavior of the momentum distribution and the deeper finite-$k$ minimum in the excitation spectrum [Fig. 5(b,c)]. This can be intuitively understood from the fact that increasing $r_0/a_0$ leads to increased screening at short range, thus reducing the exciton binding energy (dependent on the short-range behavior of the potential) relative to the bandgap renormalization (governed by the long-range part of the potential). Therefore, this suggests that TMDs could be a promising system for achieving a polariton BCS state, with the caveat that it will require a much larger electron density than in quantum wells since the excitons are more tightly bound [52, 53].

# 6 Connection to the upper polariton branch

The above discussion of the many-body properties of the light-matter coupled system focused on the ground state, i.e., the lower-polariton branch. However, the mean-field light-matter

coupled formulation also admits a second higher energy solution which, in the zero-density limit, is continuously connected to the *upper* polariton. This upper branch is typically not accessed in steady-state polariton BEC experiments, since it is metastable and far detuned in energy from the chemical potential of the lower polariton condensate. However, recent theoretical work [30, 31] has shown that the dynamical and non-equilibrium nature of the system potentially allows for a coalescence of the lower and upper branches, which preempts any crossover to the BCS regime. This lower-to-upper-branch transition provides a mechanism by which a polariton BEC can undergo a phase transition to a photon laser with increasing density, a scenario which has potentially already been realized in experiment [30]. Thus, the upper-branch solution is also physically relevant once one considers a driven dissipative system beyond thermodynamic equilibrium.

To explore this within our model, we consider a scenario where a matter bath injects electron-hole pairs into the system, while cavity photons are lost to the outside through the mirrors. This can be captured with phenomenological loss and gain rates, $\kappa$ and $\gamma$ respectively, such that Eq. (25) becomes

$$\left(\bar{\epsilon}_{\mathbf{k}} + i\gamma - \mu - 2\sum_{\mathbf{k}'} V_{\mathbf{k}-\mathbf{k}'} v_{\mathbf{k}'}^2\right) u_{\mathbf{k}} v_{\mathbf{k}} + \left[u_{\mathbf{k}}^2 - v_{\mathbf{k}}^2\right]\left(g\lambda - \sum_{\mathbf{k}'} V_{\mathbf{k}-\mathbf{k}'} u_{\mathbf{k}'} v_{\mathbf{k}'}\right) = 0, \qquad (46\text{a})$$

$$(\omega_0 - i\kappa - \mu)\lambda + g\sum_{\mathbf{k}} u_{\mathbf{k}} v_{\mathbf{k}} = 0. \qquad (46\text{b})$$

We could have equivalently obtained these equations from the Heisenberg equations of motion for the electron, hole and photon operators [54].

In the limit of low density and for sufficiently large exciton binding energy $\varepsilon_B \gg \Omega$, Eq. (46) reduces to a simple non-Hermitian Hamiltonian [55]:

$$\mu\begin{pmatrix} C \\ X \end{pmatrix} = \begin{pmatrix} \omega - i\kappa & \Omega \\ \Omega & \mu_X + i\gamma \end{pmatrix}\begin{pmatrix} C \\ X \end{pmatrix}, \qquad (47)$$

where $C$ and $X$ are the usual photon and exciton amplitudes (Hopfield coefficients), respectively, and we have the exciton chemical potential in the absence of coupling to light, $\mu_X = -\varepsilon_B + g_{xx} n_e$, with $g_{xx}$ the exciton-exciton interaction strength within the Born approximation. This non-Hermitian problem yields two eigenvalues

$$\mu_{\pm} = \frac{1}{2}\left(\lambda_X + \lambda_C \pm \sqrt{(\lambda_X - \lambda_C)^2 + 4\Omega^2}\right), \qquad (48)$$

where $\lambda_X = \mu_X + i\gamma$ and $\lambda_C = \omega - i\kappa$. These eigenvalues are in general complex, and thus in the steady state the gain rate $\gamma$ is adjusted to ensure that the chemical potential is real [31].

We can investigate the behavior beyond the low-density regime in the case of contact interactions by extending the expression (38) for the carrier density to include gain and loss rates:

$$n_e = \frac{m_r}{2\pi}\left(\mu - i\gamma + \varepsilon_B e^{\frac{\Omega^2}{(\omega_0 - \mu - i\kappa)\varepsilon_B}}\right). \qquad (49)$$

Note that this is purely phenomenological and can be viewed as an analytic continuation of the equation of state from real to complex single-particle energies. Moreover, we see that we recover the characteristic equation for the eigenvalues of Eq. (47) in the regime where $\varepsilon_B \gg \Omega$, $g_{PP} n_{tot}$.

For the equation of state, we have carrier density in terms of the (real) chemical potential, in contrast to Eq. (48), and thus we instead impose the condition that the density is real, i.e.,

$\text{Im}\{n_e\} = 0$. This yields an explicit equation for the gain parameter $\gamma$:

$$\gamma = \varepsilon_B \exp\left[\frac{\Omega^2}{(\omega_0 - \mu)^2 + \kappa^2}\frac{\omega_0 - \mu}{\varepsilon_B}\right] \times \sin\left[\frac{\Omega^2}{(\omega_0 - \mu)^2 + \kappa^2}\frac{\kappa}{\varepsilon_B}\right]. \qquad (50)$$

In the limit of large $\varepsilon_B$, where we recover the Hamiltonian in Eq. (47), this reduces to $\gamma = \kappa|C|^2/|X|^2$, which is consistent with simple models of driven-dissipative polariton condensates [56].

In Fig. 6 we show the case of very strong light-matter coupling $\Omega = 0.5\varepsilon_B$ and excitonic detuning $\delta = 0.1\varepsilon_B$, which goes beyond the regime described by the non-Hermitian Hamiltonian in Eq. (47). For a given non-zero $\kappa$, we observe that a region of densities exists where multiple steady-state solutions are possible. Here, the solid lines indicate physical solutions, where the system can exhibit hysteresis when the matter part is directly pumped. This picture is in agreement with previous results on the driven dissipative polariton system [30, 31], and also with the behavior expected in a more general theory of non-reciprocal phase transitions [57]. With increasing $\kappa$, we see that the two branches approach each other until eventually they merge at the exceptional point where $\kappa_c \simeq 0.51\varepsilon_B$ and $\mu = \omega = -0.9\varepsilon_B$. In particular, we find that the exceptional point is set by the physical cavity frequency rather than the bare frequency $\omega_0$, which is unlike the case of the singular cavity resonance in the purely equilibrium scenario.

Note that the exceptional point within our model requires a sizeable loss rate that takes us outside of the strong-coupling regime ($\Omega > \kappa_c$) in polariton experiments. Reference [30] has proposed that the exceptional point can be reached even for $\Omega > \kappa_c$ since the light-matter coupling decreases with increasing density due to Pauli blocking or phase space filling effects [2, 3, 58]. However, in our fully renormalized theory, we observe no such decrease of the Rabi splitting with increasing density (see, e.g., the $\kappa = 0$ lines in Fig. 6) even though the BCS wave function (21) clearly contains Pauli blocking. Therefore, the present BCS mean-field theory does not capture the loss of strong coupling observed in experiment at large densities, and we possibly require additional many-body effects beyond phase space filling in order to theoretically describe a density-driven exceptional point in realistic experiments.

# 7 Concluding remarks

In summary, we have determined the many-body properties of exciton-polariton condensates within a BCS variational theory. We have performed calculations for a variety of different interactions between charge carriers (both long-range and contact) within a fully renormalized approach, where the results obtained are independent of any UV cutoff in the low-energy microscopic model. In particular, we have demonstrated that the BCS theory recovers the single-polariton properties in the zero-density limit, and the Born approximation of polariton scattering at low density. At higher densities, we have discussed how a photon resonance tends to confine the ground state to negative chemical potential, which poses a challenge to observing the BCS regime. Here, we found that TMD monolayers appear to be particularly promising for achieving the BCS limit, due to the nature of the carrier interactions in these materials.

The mean-field theory studied in this work can be extended to consider the effects of dynamics and quantum fluctuations. An interesting question in this context is the potential connection of the upper and lower many-body branches which, due to the dynamical and non-equilibrium nature of exciton-polariton condensates, can coalesce at an exceptional point where loss, gain, and Rabi coupling are equal.

An especially interesting extension of our work is to investigate the role of spin (photon polarization), since interactions between polaritons of opposite spin can be strongly enhanced

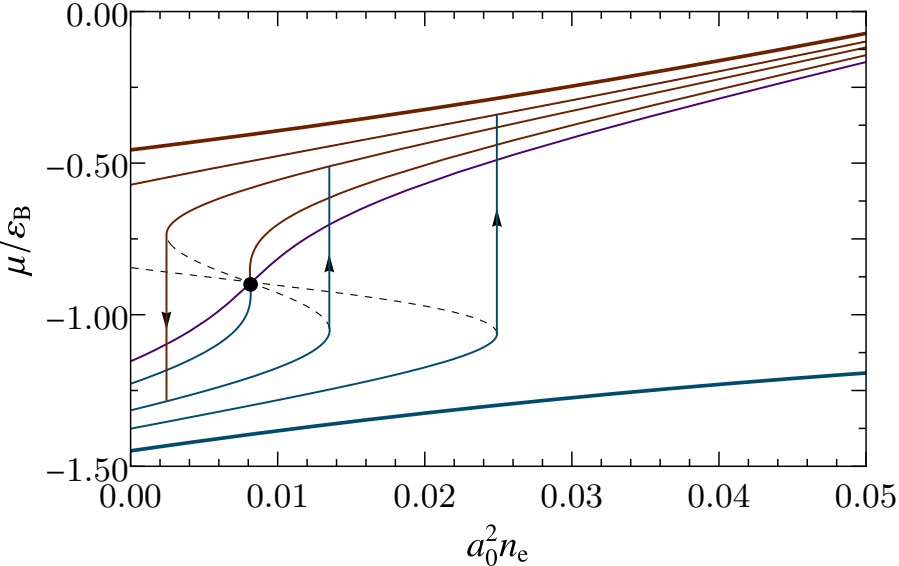

Figure 6: The chemical potential for the upper (red) and lower (blue) branches in the driven-dissipative polariton system. The results are obtained by solving Eqs. (49) and (50) for fixed Rabi coupling $\Omega = 0.5\varepsilon_B$, detuning $\delta = 0.1\varepsilon_B$, and for loss rates: $\kappa = 0$, $0.3\varepsilon_B$, $0.4\varepsilon_B$, $0.51\varepsilon_B$ and $\kappa = 0.6\varepsilon_B$ (top to bottom for the upper branch, and vice versa for the lower branch). Since the matter component is assumed to be pumped directly, the electron density $n_e$ is proportional to the pump power [31]. The black dot indicates the exceptional point and the dashed lines represent unstable solutions. The arrows indicate the hysteresis loops obtained by changing the pump power in both directions.

when their collision energy is close to a biexciton resonance [59, 60]. In the vicinity of such a resonance, the polariton interaction is strongly energy dependent, and it may therefore be possible to effectively tune to resonance using the many-body energy shifts shown in Figs. 3 and 4. As a result, the BEC-BCS crossover and associated phases such as photon lasing can become strongly dependent on polarization. This could have implications for polariton condensation in TMDs, where the polarization has already been shown to have a strong effect on the interactions [61].

# Acknowledgments

We thank Elena Ostrovskaya, Eliezer Estrecho, Emma Laird, and Guangyao Li for useful discussions.

**Funding information** We acknowledge support from the Australian Research Council Centre of Excellence in Future Low-Energy Electronics Technologies (CE170100039). JL and MMP are furthermore supported through the Australian Research Council Future Fellowships FT160100244 and FT200100619, respectively.

## A   Low-density regime

In the limit of vanishing density $n_{\text{tot}} \to 0$, we recover the behavior of a single polariton of energy $E$ in Eq. (9), where $\mu \approx E$, $\lambda \approx \sqrt{n_{\text{tot}}}\gamma$ and $\nu_{\mathbf{k}} \approx \sqrt{n_{\text{tot}}}\varphi_{\mathbf{k}}$ (see Sec. 3). To obtain the leading order correction $\delta\mu$ to the chemical potential due to the interactions between polaritons, we expand Eq. (25) in powers of $n_{\text{tot}}$, keeping only the lowest order terms,

$$\left(\bar{\epsilon}_{\mathbf{k}} - \mu - 2n_{\text{tot}}\sum_{\mathbf{k}'}V_{\mathbf{k}-\mathbf{k}'}\varphi_{\mathbf{k}'}^2\right)\varphi_{\mathbf{k}}u_{\mathbf{k}} + \left[1 - 2n_{\text{tot}}\varphi_{\mathbf{k}}^2\right]\left(g\gamma - \sum_{\mathbf{k}'}V_{\mathbf{k}-\mathbf{k}'}\varphi_{\mathbf{k}'}u_{\mathbf{k}'}\right) = 0, \qquad \text{(A.1a)}$$

$$(\omega_0 - \mu)\gamma + g\sum_{\mathbf{k}}\varphi_{\mathbf{k}}u_{\mathbf{k}} = 0. \qquad \text{(A.1b)}$$

Here we have divided out $\sqrt{n_{\text{tot}}}$ and we have used the condition $u_{\mathbf{k}}^2 + v_{\mathbf{k}}^2 = 1$. The latter also allows us to expand the parameter $u_{\mathbf{k}}$ in terms of density: $u_{\mathbf{k}} \approx 1 - \frac{n_{\text{tot}}}{2}\varphi_{\mathbf{k}}^2$.

To proceed, we make use of the normalization of the polariton state by multiplying Eq. (A.1a) by $\varphi_{\mathbf{k}}$ and summing over momentum $\mathbf{k}$. Then, using $\mu = E + \delta\mu$ and Eq. (9), we obtain the low-density expressions

$$-\delta\mu\sum_{\mathbf{k}}\varphi_{\mathbf{k}}^2 - \frac{n_{\text{tot}}}{2}\sum_{\mathbf{k}}\varphi_{\mathbf{k}}^3\Big[\overbrace{(\bar{\epsilon}_{\mathbf{k}} - E)\varphi_{\mathbf{k}} - \sum_{\mathbf{k}'}V_{\mathbf{k}-\mathbf{k}'}\varphi_{\mathbf{k}'}}^{-g\gamma}\Big]$$
$$+ 2n_{\text{tot}}\left[\sum_{\mathbf{k},\mathbf{k}'}V_{\mathbf{k}-\mathbf{k}'}\left(\varphi_{\mathbf{k}}^3\varphi_{\mathbf{k}'} - \varphi_{\mathbf{k}}^2\varphi_{\mathbf{k}'}^2\right) - g\gamma\sum_{\mathbf{k}}\varphi_{\mathbf{k}}^3\right] = 0, \qquad \text{(A.2a)}$$

$$-\delta\mu\gamma - \frac{n_{\text{tot}}}{2}g\sum_{\mathbf{k}}\varphi_{\mathbf{k}}^3 = 0, \qquad \text{(A.2b)}$$

where we have used the symmetry $V_{\mathbf{k}-\mathbf{k}'} = V_{\mathbf{k}'-\mathbf{k}}$ in the first line. Combining the equations gives

$$-\delta\mu\Big[\sum_{\mathbf{k}}\varphi_{\mathbf{k}}^2 + \gamma^2\Big] - 2n_{\text{tot}}g\gamma\sum_{\mathbf{k}}\varphi_{\mathbf{k}}^3 + 2n_{\text{tot}}\sum_{\mathbf{k},\mathbf{k}'}V_{\mathbf{k}-\mathbf{k}'}\left(\varphi_{\mathbf{k}}^3\varphi_{\mathbf{k}'} - \varphi_{\mathbf{k}}^2\varphi_{\mathbf{k}'}^2\right) = 0. \qquad \text{(A.3)}$$

Since the polariton state is normalized, we finally have

$$\delta\mu = 2\left[\sum_{\mathbf{k}}(\bar{\epsilon}_{\mathbf{k}} - E)\varphi_{\mathbf{k}}^4 - \sum_{\mathbf{k},\mathbf{k}'}V_{\mathbf{k}-\mathbf{k}'}\varphi_{\mathbf{k}}^2\varphi_{\mathbf{k}'}^2\right]n_{\text{tot}}, \qquad \text{(A.4)}$$

where we used Eq. (9) to rewrite the $\gamma$ term in Eq. (A.3). Equation (A.4) corresponds to $\delta\mu = g_{\text{PP}}n_{\text{tot}}$ with $g_{\text{PP}}$ given by Eq. (42) once we take the solution for the lower polariton: $E \to E_-$ and $\varphi_{\mathbf{k}} \to \varphi_{\text{LP}\mathbf{k}}$. Thus, we have shown that the BCS approach describes the polariton-polariton interactions within the Born approximation.

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
