# Peer review of "Quasi-equilibrium polariton condensates in the non-linear regime and beyond"

_SciPost Physics, doi:SciPost Phys. 15, 116 (2023)_

## Round 1 · Referee Report · Ryo Hanai (Referee 1) · 2023-3-16

Strengths

- The authors were able to formulate a meanfield theory for electron-hole photon condensate without introducing UV cutoff.
- This formalism enabled the authors to show that the BCS regime is enhanced for Rytova-Keldysh potential compared to Coulomb or contact interaction.

Weaknesses

- The overall picture of what the authors obtained is not very different from the known results.
- The treatment of dissipation is problematic even at the phenomenological level.

Report

In this work, the authors present a mean-field theory of an electron-hole-photon mixture in a two-dimensional semiconductor microcavity. While there have been several works that performed a similar analysis to the model considered in this manuscript, the novelty here is that they have employed an appropriate renormalization scheme. They considered three types of interactions: Coulomb interaction, contact interaction, and Rytova-Keldysh potential, and found that the Rytova-Keldysh potential offers the largest parameter regime of the BCS regime. This treatment allowed the authors to eliminate the ultraviolet cutoff dependence from the model.
The results are non-trivial, and the manuscript is well-written. I, therefore, recommend its publication in SciPost Physics once the following comments are addressed.

Requested changes

1. Is there an intuitive understanding of why the Rytova-Keldysh potential provides a more significant BCS regime than the Coulomb interaction? I would have thought that the shorter range nature of the Rytova-Keldysh potential would suppress the band renormalization that made it possible to exhibit the BCS regime.

2. In Sec. IV, the authors consider the effect of dissipation by introducing a gain and a loss to the exciton and photon component, respectively, in a phenomenological manner. However, I find some of their treatment problematic.

The chemical potential μ must not have an imaginary part since its presence implies damping or gain of the condensate. Note that, in the canonical ensemble, the condensate has an oscillating phase determined by the chemical potential, as Delta(t) = Delta_0 exp[I 2μt]; see Ref. [49]. This contradicts the assumption that the system is in a steady-state. In general, there would be a nonlinear imaginary term that gives rise to the saturation effect (as done in [Wouters and Carusotto, PRL 99, 140402 (2007)] for the one-component case and Ref. [49] and [Hanai and Littlewood, PRR 2, 033018 (2020)] for the two-component case) that automatically makes the steady-state condition (=μ being real) satisfied in the long-time limit.
This can be achieved by interpreting their gamma (RX in Ref. [49]) to include these nonlinear effects. In particular, gamma should be considered as a parameter that is determined by demanding μ to be real, as done in Ref. [49] (instead of assuming gamma = kappa as done in the manuscript).

On the other hand, there is no need to add an imaginary part to the electron density Eq. (37). In the Keldysh formalism, adding dissipation to the system would make the spectrum have a Lorentz distribution rather than the delta-function of the mean-field approximation but would never give rise to an imaginary part of the density (See, e.g., Ref. [14].). Therefore, solving Eq. (49) is not necessary. What one should solve instead is the requirement that the chemical potential μ is real, WITHOUT assuming by hand that gamma = kappa, as mentioned above.

One of the reasons that I strongly recommend the authors to perform the above analysis is that assuming gamma = kappa would lead to a somewhat misleading conclusion that one always goes through an exceptional point by tuning the density. Instead, as pointed out in Ref. [49] (and more recently in [Fruchart, Hanai, Littlewood, Vitelli, Nature 592, 363 (2021)] from a symmetry perspective), one needs to fine-tune TWO parameters to go through an exceptional point in a U(1)-broken system like exciton-polariton condensates.

3. I am confused by the authors’ comment “Reference [49] has proposed that … due to Pauli blocking or phase space filling effects [2, 3, 54]. However, we observe no such decrease of the Rabi splitting with increasing density ...” I am pretty sure that the Pauli blocking effect already appears at a meanfield level; see Eq. (S81) of the SI of Ref. [49]. Could the authors comment on why they could not see such effects in more detail?

  • validity: high
  • significance: ok
  • originality: good
  • clarity: high
  • formatting: excellent
  • grammar: excellent

Author:  Brendan Mulkerin  on 2023-05-24  [id 3682]

(in reply to Report 1 by Ryo Hanai on 2023-03-16)

RESPONSE TO REFEREE:
We thank the Referee for his comments and detailed analysis of our manuscript and we appreciate his supportive report and recommendations.

We address his comments point by point in the attached document, which includes changes to the manuscript highlighted in blue.

Attachment:

respone_final_Ryo.pdf

---

## Round 2 · Referee Report · Ryo Hanai · 2023-7-6

Report

I am happy with all of the authors’ comments. The mechanism of how the BCS regime enhances in the Rytova-Keldysh potential case by the enhanced band-renormalization due to the screening of the short-range part is interesting. It was a failure on my part to miss some of the qualitative novelty of the present work.
I am happy to recommend the publication of the manuscript to SciPost in its current form.

---

## Round 2 · Author Response

To: the Editorial Office,
SciPost
Quasi-equilibrium polariton condensates in the non-linear regime and beyond
by Ned Goodman, Brendan C. Mulkerin, Jesper Levinsen, Meera M. Parish
May 24, 2023

We thank the Referee and Editor for their insightful comments, which have helped us to
improve our manuscript. After revising the manuscript to take into account the comments,
we would like to resubmit for publication in SciPost.

Sincerely,
Ned Goodman,
Brendan Mulkerin,
Jesper Levinsen,
Meera Parish

---

## Round 2 · List of Changes

RESPONSE TO REFEREE:
We thank the Referee for his comments and detailed analysis of our manuscript and we appreciate his supportive report and recommendations.

Below we address his comments point by point.
1. Is there an intuitive understanding of why the Rytova-Keldysh potential provides a more significant BCS regime than the Coulomb interaction? I would have thoughtthat the shorter range nature of the Rytova-Keldysh potential would suppress the band
renormalization that made it possible to exhibit the BCS regime.

The Rytova-Keldysh potential is actually effectively more long-ranged than the un screened Coulomb interaction: it is screened at short distances by the dielectric environment rather than at long distances like in the usual case of Thomas-Fermi screening, and it thus retains the same 1/r behavior at large r while having a reduced potential at small r. In particular, this means that the bandgap renormalization (which is governed by the long-range part of the potential) will be enhanced relative to the exciton binding energy (which is affected by the short-distance part), thus making it easier to access the BCS regime.
We have now added the following sentence to highlight the above point:
“This can be intuitively understood from the fact that increasing r0/a0 leads to increased screening at short range, thus reducing the exciton binding energy (dependent on the short-range behavior of the potential) relative to the bandgap renormalization (governed by thelong-range part of the potential).”

2. In Sec. IV, the authors consider the effect of dissipation by introducing a gain and a loss to the exciton and photon component, respectively, in a phenomenological manner. However, I find some of their treatment problematic. The chemical potential µ must not have an imaginary part since its presence implies damping or gain of the condensate. Note that, in the canonical ensemble, the condensate has an oscillating phase determined by the chemical potential, as ∆(t) = ∆0exp[I2µt]; see Ref. [49]. This contradicts the assumption that the system is in a steady-state. In general, there would be a nonlinear imaginary term that gives rise to the saturation effect (as done in [Wouters and Carusotto, PRL 99, 140402 (2007)] for the one-component case and Ref. [49] and [Hanai and Littlewood, PRR 2, 033018 (2020)] for the two-component case) that automatically makes the steady-state condition (= µ being real) satisfied in the long-time limit. This can be achieved by interpreting their gamma (RX in Ref.[49]) to include these nonlinear effects. In particular, gamma should be considered as a parameter that is determined by demanding µ to be real, as done in Ref. [49] (instead of assuming gamma = kappa as done in the manuscript).

We thank the Referee for alerting us to this issue, which was very much an oversight on our part. We certainly agree that the chemical potential of the condensate needs to be real in order to be in the steady state. We have now revised our analysis of the upper and lower
branches with gain/loss, where we have allowed γ to vary such that both µ and the density are real. We have also cited [Hanai and Littlewood, PRR 2020] as the new reference [31].

On the other hand, there is no need to add an imaginary part to the electron density Eq. (37). In the Keldysh formalism, adding dissipation to the system would make the spectrum have a Lorentz distribution rather than the delta-function of the mean-field approximation but would never give rise to an imaginary part of the density (See, e.g., Ref. [14].). Therefore, solving Eq. (49) is not necessary. What one should solve instead is the requirement that the chemical potential µ is real, WITHOUT assuming by hand that γ = κ, as mentioned above.

We needed to solve Eq. (49) [Eq. (50) in the revised version] to ensure that the density remained real. The reason is that our analytic expression for the equation of state has the density as a function of chemical potential, in contrast to the 2×2 matrix at low density in Eq. (47) where we instead have the chemical potential as a function of density. Thus, we have a requirement on the density rather than µ, which we can take to be real at the outset.
However, as noted above, we agree that we should not have assumed γ = κ, so we have now modified the equations to reflect this. Indeed, one can show that |X|^2 γ = |C|^2 κ for Eq. (47) in the steady state and we have checked that our analysis agrees with this in the limit of low density (large exciton binding energy εB).

One of the reasons that I strongly recommend the authors to perform the above analysis is that assuming gamma = kappa would lead to a somewhat misleading conclusion that one always goes through an exceptional point by tuning the density. Instead, as pointed out in Ref. [30] (and more recently in [Fruchart,Hanai, Littlewood, Vitelli, Nature 592, 363 (2021)] from a symmetry perspective), one needs to fine-tune TWO parameters to go through an exceptional point in a U(1)-broken system like exciton-polariton condensates.

We thank the referee for bringing the reference Fruchart et al. to our attention, which is now included in the manuscript as the new reference [59].

3. I am confused by the authors’ comment “Reference [49] has proposed that ... due to Pauli blocking or phase space filling effects [2, 3, 54]. However, we observe no such decrease of the Rabi splitting with increasing density ...” I am pretty sure that the Pauli blocking effect already appears at a meanfield level; see Eq. (S81) of the SI of Ref. [49]. Could the authors comment on why they could not see such effects in more detail?

Of course, we agree with the Referee that Pauli blocking is present in the BCS formalism, and indeed we have the same Pauli blocking term as in Eq. (S81) of Ref. [30] ([49] in the previous version) in our Eq. (25a). What we meant was simply that we do not observe
a closing of the Rabi splitting in any of our calculations, numerical or analytical. The behavior of the Rabi splitting involves an interplay between photon-mediated electron-hole interactions and Pauli blocking, rather than just Pauli blocking alone, and it is crucial to use a properly renormalized theory to describe this. For instance, previous calculations that ignore the underlying UV divergence have hugely overestimated the role of saturation in the polariton-polariton interaction strength (see the discussion in Ref. [26]). We have now tried to make this point clearer in the manuscript.

Finally, we would like to respond to the Referee’s assessment:
The overall picture of what the authors obtained is not very different from the known results.

We respectfully disagree with this assessment. For instance, we have demonstrated that the properly renormalized theory has very small saturation of the Rabi coupling. This is as opposed to Ref. [22], which predicted a strongly varying Rabi coupling with detuning, which we believe arises from the unrenormalized model used in that work (i.e., the detuning did not correspond to that which would be found in solving the single-polariton problem). This in turn led to a disagreement in that work between the observed low-energy behavior of the chemical potential and that expected based on the two-body polariton-polariton scattering [Eq. (41)], i.e., they did not recover the expected polariton energy at zero density, and the polariton interactions weakened more rapidly with decreasing detuning than would be expected from the Born approximation. Conversely, our fully renormalized theory is the first to demonstrate how the correct low-energy behavior of the chemical potential arises from the BCS equations.
The heart of the matter is the correct identification of the two-body physics, which provides the starting point of many-body physics. This can also be illustrated by how Ref. [11] found that the excitons become Frenkel-like in the photon dominated regime (strongly bound with a small radius). Instead, in our fully renormalized model we find that the electron-hole separation approaches a universal value, independent of the details of the interactions.

RESPONSE TO EDITOR REPORT

We thank the Editor for their time in providing us with valuable comments and a thorough analysis of our manuscript. We also appreciate their supportive report and recommendations. Below we address their comments point by point.

1. On a more global scale, I would like to see better explanations of the various quantities introduced, and better identification of what is new and what is taken from previous works. For instance, I believe Sec II is not new, while new results start in Sec III; is
this correct? Sec II: How realistic is the model?

The model (1) is standard in the literature of semiconductors strongly coupled to light, and relies on the assumption that we are close enough to the band edge that we are able to describe the electrons and holes as having quadratic dispersions. This is realistic as long as the interparticle separation between electrons (or holes) greatly exceeds the lattice spacing, which is generally the case in these systems. However, it was only recently realized that this model needs to be renormalized, since it has an ultraviolet divergence. As shown, for instance, in the new reference [36], the correctly renormalized model can accurately describe recent experimental measurements of exciton-polaritons in the presence of both strong light-matter coupling and strong magnetic fields (new reference [35]), where previous perturbative methods fail.
We have now amended the manuscript throughout to include more descriptions of the quantities used in the calculations, and we have been more explicit about how the model introduced in Sec II is not new. The latter part of Sec II is a summary of previous works
for the reader to understand the renormalization scheme for each interaction, since renormalization is a key aspect of our work. On the other hand, Sec III signals the beginning of our original results, and we have highlighted this.

2. It looks like there is a lot of freedom e.g. in the choice of the potential. What is the “true” potential? What are the results / proposals on this?

The choice of the potential depends on the precise geometry under investigation. In the case of quantum wells of finite width, the dielectric environment is approximately uniform, and hence the effective interaction between electrons and holes is simply the usual Coulomb interaction ∼ −1/r. For atomically thin semiconductors, the dielectric constant varies strongly between the thin layer and the surrounding medium. This leads to an effective geometric screening of the Coulomb interaction, resulting in the so-called Rytova-Keldysh potential. Our model applies to both of those scenarios, which are both experimentally realistic. On the other hand, the highly screened contact interaction provides an important benchmark for other theories as it allows us to derive semi-analytic results. It furthermore allows us to estimate the effects of screening on our results.
We have now modified the discussion of the different potentials on page 2 to address this point.

3. At the beginning of sect II it would be good already to have the Rytova-Keldysh potential, and the long-range potentials; eq (11) and (12) should be earlier, to help the reader follow the discussion.

We thank the Editor for their suggestion and we have moved Eqs. (11-12) to after Eq. (4) to help the reader follow the discussion.

4. Sect II.1: I was confused as to where the long-range interaction was supposed to be. Is this in the definition of the potential? I understand it means that Vq is supported on small q’s only, in some way, but this should be made clearer already.

Following the above comment, we have now made this section clearer by introducing the long-range Coulomb and Rytova-Keldysh interactions straight after Eq. (4). We have also explicitly stated that both these potentials scale as 1/r at large interparticle separation.

5. “the lowest energy 1s state” what does 1s mean? Can the lowest-energy state be made more explicit? “Projecting the Schr¨odinger” Is there a need to project? This is just obtained by applying H from (1) to the state (5) I believe.

The lowest energy 1s state is the ground state in the s-wave channel. We have now made this clearer in the text. By “projecting” we mean that the two parts of Eq. (9) of the new manuscript correspond to ⟨0| e_{k}h_{−k}( H − E) |Ψ⟩ = 0 and ⟨0| c_0( H − E) |Ψ⟩ = 0, respectively, as now stated above Eq. (9).

6. “The Hopfield coefficients...” in this sentence: what are Hopfield coefficients? What is C± and X±? (That is, how are they defined? There is a partial definition later in eq 46 but this should be clarified much earlier.)

The Hopfield coefficients are the amplitudes of the matter and photon parts of the polariton wavefunction, the modulus square of which gives the excitonic and photonic fractions. Since in our ansatz γ is the photonic contribution to the wavefunction we can directly identify this with the photonic Hopfield coefficient C±. Using the normalization condition, we can then find the matter component X±. We have made the correspondence between the photonic part of the ansatz, γ, the Hopfield coefficients, and the relationship between X and C explicit.

7. I suppose γ± is the value of γ for the two branches E±. But it would be clearer to make more explicit how to go from (7) to the form (2) of E±.

Within our model, we must solve the Schr¨odinger equation (9) (or equivalently Eq. (10) which was (7) in the previous version) using the identification of light-matter coupling parameters in Eqs. (12, 13). At weak light-matter coupling (Ω ≪ εB), this was found to
match well with Eq. (2) in Ref. [26], while at strong light-matter coupling there starts to be a deviation (for the coupling strengths considered in this work, this difference is very small).
We have now made this procedure more explicit.

8. Sec II.2: is it physically sensible to use the same cutoff for the short-range interaction of particle-holes and for the coupling to light? Do we expect results to be independent of how these cutoffs are chosen? Perhaps a short discussion of this should be added.
Both cutoffs are related to the behavior of the electron-hole wave function at short distances.

This is in turn related to the lattice spacing of the underlying system, and hence both cutoffs are expected to be approximately the inverse lattice spacing. In practice, the low-energy physics that we describe is independent of the precise manner in which the
renormalization is carried out, and therefore our results are independent of this choice.
We have now included this discussion on page 4.

9. “In the absence of light-matter interactions...” here it would be good to clarify where/ how ϵB appears in (4).

We have included an additional equation with the bound state definition from Eq. (4), see Eq. (14) in the revised manuscript.

10. “Moreover, unlike in the Coulomb case, there is only the single 1s electron-hole bound state” I don’t understand this sentence - please add explanations.

The number of bound electron-hole states depends on the potential chosen, and for the contact interactions there is only a single bound state. We have now rephrased the corresponding paragraph to avoid confusion.

11. Sec III: Eq 31: where does Ek come from? I followed all calculations until there - how is Ek defined?

We thank the referee for bringing up this point, which made us realise that our wording was rather imprecise since the Ek defined in the text is actually half the pair-breaking energy, i.e., the average energy of two quasiparticles. We have now been explicit about how Ek is calculated at the end of Sec III.

12. Sec V: Eq 40: has gPP been defined, besides the words here and that equation? Is that equation its definition? I believe eq 40 is its definition (from my reading of Appendix A); please make it clear.

In the low-density limit, we have a weakly interacting BEC of polaritons. In this case, the polariton chemical potential has two contributions: Its zero-density energy E− and the interaction-induced mean-field shift due to the remaining polaritons gppntot. Equation (41) [(40) in the previous version] defines the interaction constant for polariton-polariton interactions gpp, which must be calculated within the model for a given electron-hole interaction potential. In our work, we show that the BCS ansatz allows us to calculate gpp within the Born approximation, Eq. (42).
We have now included a reference to a standard textbook which has a detailed discussion of BECs in the low-density limit.

13. At the beginning of sec V: it would be nice to have a discussion of how the BEC-BCS crossover is actually defined, in terms of the equations / parameters discussed until there, and what is the generally expected physics. There is of course elements of such
discussion afterwards, but it would be good to have it already from the beginning. For instance, to me the pair-size measure is perhaps the clearest.

We thank the Editor for their comment. In order to clarify the meaning of the BEC-BCS crossover we have rewritten the opening paragraph of this section to describe the two limits. We have also explicitly referred to the equations that we use to obtain our results.
14. Sub-divisions: the first bit of sec V only discusses Coulomb and contact interactions, while then a following sub-section discusses RK. It would be good to make sub-sections clearer to account for this.
We have now introduced new subsection headings in Sec V.

You are currently on this page

Resubmission 2211.03321v2 on 26 May 2023

---

## Editorial Decision

published